# Genetic Predictors of Psychosomatic Symptoms in Individuals with Metabolic Syndrome: Insights from a Cross-Sectional Study in Kazakhstan

**DOI:** 10.3390/ijerph22121853

**Published:** 2025-12-12

**Authors:** Dinara Nemetova, Mira Zhunissova, Gulnaz Nuskabayeva, Ugilzhan Tatykayeva, Ainash Oshibayeva, Nursultan Nurdinov, Dilbar Aidarbekova, Ainur Turmanbayeva, Yerbolat Saruarov, Zhanar Zhagiparova, Yusuf Ozkul, Nuriye Gokce, Karlygash Sadykova

**Affiliations:** 1Department of Special Clinical Disciplines, Faculty of Medicine, Khoja Akhmet Yassawi International Kazakh-Turkish University, Turkestan 161200, Kazakhstan; nemetova.dinara@ayu.edu.kz (D.N.); mira.zhunissova@ayu.edu.kz (M.Z.); nuskabayeva.gulnaz@ayu.edu.kz (G.N.); dilbar.aidarbekova@ayu.edu.kz (D.A.); ainur.turmanbaeva@ayu.edu.kz (A.T.); yerbolat.saruarov@ayu.edu.kz (Y.S.); 2Department of Fundamental Medical Sciences, Faculty of Dentistry, Khoja Akhmet Yassawi International Kazakh-Turkish University, Turkestan 161200, Kazakhstan; ugilzhan.tatykayeva@ayu.edu.kz (U.T.); nursultan.nurdinov@ayu.edu.kz (N.N.); 3Department of Public Health and Research, Faculty of Medicine, Khoja Akhmet Yassawi International Kazakh-Turkish University, Turkestan 161200, Kazakhstan; ainash.oshibayeva@ayu.edu.kz; 4Faculty of Postgraduate Higher Medical Education, Khoja Akhmet Yassawi International Kazakh-Turkish University, Turkestan 161200, Kazakhstan; zhanar.zhagiparova@ayu.edu.kz; 5Department of Medical Genetics, School of Medicine, Erciyes University, Kayseri 38039, Turkey; ozkul@erciyes.edu.tr (Y.O.); nuriyecoskun@erciyes.edu.tr (N.G.)

**Keywords:** metabolic syndrome, psychosomatic disorders, DASS-21, genetic predictors, cross-sectional study

## Abstract

Background: Metabolic syndrome (MetS), a cluster of cardiometabolic abnormalities including elevated blood pressure, impaired glucose regulation, dyslipidemia, and increased waist circumference is increasingly recognized as a condition linked to both physical and psychological health risks. This study aims to investigate genotype-specific differences in psychological distress between healthy individuals and those with metabolic disorders, as well as to examine potential gene metabolic status interactions. Methods: This study is a cross-sectional analysis conducted in Turkistan city in the Southern region of Kazakhstan. Participants (healthy and those with metabolic syndrome) were invited to take part in the study by random sampling from the Khoja Akhmet Yassawi Kazakh-Turkish International University Medical Center. Consenting individuals provided a genetic analysis. Psychosomatic indicators were assessed using the Perceived Stress Questionnaire (PSQ) and the Depression, Anxiety, and Stress Scale (DASS-21). Results: A total of 200 individuals participated, with an approximately 3:1 ratio of women to men. The mean age in years was 50.4 ± 9.5 and 48.8 ± 7.7 for men and women, respectively. Preliminary analyses showed variations in cognitive and psychosomatic measures among individuals with metabolic syndrome, but no associations with genetic variants, and no significant group differences across key psychosomatic indicators when stratified by metabolic or genetic factors. However, a significant difference in LPL-Anxiety between genotypes GA-GG (*p* < 0.05) was found. Conclusions: Variations in metabolic and genetic factors within the studied population were not associated with measurable differences in stress or depressive symptoms.

## 1. Introduction

There is a continual increase in the worldwide prevalence of metabolic disorders, representing a major contributor to the development of diabetes mellitus, cardiovascular diseases, cancers, and premature deaths [1,2]. In particular, abdominal obesity contributes to the development of metabolic syndrome (MetS) by causing imbalances in fatty acids, insulin resistance, and other metabolic disruptions [3,4]. MetS is associated with a higher prevalence and greater severity of mood and anxiety disorders [5,6,7]. Compared to metabolically healthy individuals, people with MetS have a two-fold increased risk of mental health disorders, including anxiety, depression, and stress [6].

In addition to environmental and lifestyle factors, genetic predisposition plays a significant role in metabolic and emotional health, with variations in genes regulating lipid metabolism and neuroendocrine function found to influence the susceptibility to metabolic disorders and affect emotional regulation [8]. For instance, the lipoprotein lipase (*LPL*) gene encodes a key enzyme that hydrolyzes triglycerides in circulating chylomicrons and very low-density lipoproteins, releasing free fatty acids for tissue uptake [9,10]. Common *LPL* polymorphisms, such as the *PvuII* (rs285) and *S447X* (rs328) variants, have been associated with altered lipid profiles and risk of MetS [11,12]. For example, the *PvuII P^+^* allele has been linked to increased triglyceride levels and higher metabolic risk, whereas the *S447X G* allele often shows a protective lipid-lowering effect [9,12]. Although *LPL* is not directly implicated in mood regulation, its influence on metabolic health could indirectly affect mental well-being through inflammation, insulin signaling, and neurovascular changes [12,13].

Another gene is the neuropeptide Y gene (NPY), which plays a direct emotional regulating role, and is one of the most abundant neuropeptides in the central nervous system that influences feeding behavior, circadian rhythms, stress responses, and mood [14]. The gene is thought to exert anxiolytic effects via receptor-mediated inhibition of the hypothalamic–pituitary–adrenal (HPA) axis and modulation of limbic brain circuits [15]. Low *NPY* expression has been associated with heightened emotional reactivity, increased anxiety, and greater vulnerability to depressive disorders [16]. Functional variant rs16139 (Leu7Pro) was reported to modify peptide release and has been associated with mood disorders in some populations [17].

However, the link between MetS and mental health may be mediated through systemic inflammation, endothelial impairment, and neurotransmitter imbalances [18]. For example, a metabolically unhealthy individual with a low *NPY* genotype expression may be particularly vulnerable to stress-related disorders [19]. Similarly, *LPL* variants that predispose to dyslipidemia could exacerbate neurovascular changes that contribute to depression [18,19]. Therefore, the current study aims to investigate genotype-specific differences in psychological distress between metabolically healthy and unhealthy subgroups and examine potential gene metabolic status interactions to understand the link between MetS and mental health status.

## 2. Methods

### 2.1. Study Design

This cross-sectional study included adults aged ≥ 18 years, recruited from community health centers in Turkistan City and Clinics of Khoja Akhmet Yassawi International Turkish-Kazakh University. Peripheral venous blood samples were collected under sterile conditions and stored at −20 °C until analysis. Participants were classified as metabolically healthy or unhealthy as diagnosed by health professionals. For example, individuals were considered metabolically healthy if they exhibited normal blood pressure, fasting glucose, triglycerides, HDL-cholesterol levels, and waist circumference, with no clinical diagnosis of metabolic syndrome components. Conversely, participants were classified as metabolically unhealthy if they had elevated blood pressure, impaired fasting glucose or diagnosed diabetes, elevated triglycerides, reduced HDL-cholesterol, or increased waist circumference consistent with central obesity. Inclusion criteria: adults (males and females) aged ≥ 18 years who provided informed consent and had complete clinical and metabolic assessments. Exclusion criteria: individuals with major psychiatric disorders under active treatment, chronic neurological diseases, or severe acute illnesses. Sociodemographic characteristics, including age, sex, education level, occupation, and lifestyle factors, were collected from all participants to contextualize the findings of the cross-sectional study.

### 2.2. Ethical Approval

The study protocol was approved by the Institutional Research Ethics Committee at Khoja Akhmet Yassawi International Turkish-Kazakh University Ethics Committee (30.05.2024, №30). All procedures and methods were conducted in accordance with the ethical principles of the Declaration of Helsinki. All participants provided written informed consent by completing a voluntary consent form before the study.

### 2.3. Psychometric Analysis

Participants’ levels of anxiety, depression, and stress were measured using the Depression, Anxiety, and Stress Scale-21 Items (DASS-21), a self-administered questionnaire consisting of three separate scales, each containing seven items grouped into subscales with related content. Sample items include: “I couldn’t seem to experience any positive feeling at all” (depression), “I experienced trembling (e.g., in the hands)” (anxiety), and “I felt that I was using a lot of nervous energy” (stress). Participants rated each item based on their emotional experiences during the past week, using a scale from 0 (did not apply to me at all) to 3 (applied to me very much). In this study, depression, anxiety, and stress were treated as ordinal variables. Conventional cut-off scores for severity were applied as follows: Depression—normal: 0–9, mild: 10–13, moderate: 14–20, severe: ≥21; Anxiety—normal: 0–7, mild: 8–9, moderate: 10–14, severe: ≥15; Stress—normal: 0–14, mild: 15–18, moderate: 19–25, severe: ≥26. As per the DASS-21 scoring method, scores for each scale were multiplied by two, yielding a maximum possible score of 42 per scale [20]. The psychological stress levels were assessed using the Perceived Stress Questionnaire (PSQ), a questionnaire designed to measure stress over the past month. The PSQ consists of 30 items rated on a 4-point Likert scale ranging from “almost never” to “usually.” It also captures multiple domains of perceived stress, including tension, worry, and demands, which help in providing a comprehensive evaluation of cognitive and emotional stress.

### 2.4. Study Validation and Reproducibility

Both the DASS-21 and PSQ used in this study had previously undergone validation among Kazakhstani populations, as reported by Uristemova et al. (2023) [21]. To further ensure the accuracy and suitability of the tools, as well as their reliability within our target sample, we conducted a small pilot test with ten randomly selected participants. Feedback from this pilot was used solely to refine the survey administration process; the pilot data itself was excluded from the final analytical dataset.

### 2.5. Genotyping

PCR amplifications were carried out according to the manufacturer’s guidelines as described by Doan et al., 2023 [22]. Briefly, each reaction had a volume of 25 µL, containing 5 µL of nuclease-free water, 4 µL of 10× LA Buffer II (Mg^2+^ plus), 4 µL of dNTP mix (2.5 mM each), 0.5 µL of forward primer (10 pmol/µL), 0.5 µL of reverse primer (10 pmol/µL), 0.5 µL of Takara LA Taq DNA polymerase (5 U/µL), 5.5 µL of betaine (5 M), and 5 µL of template DNA (200 ng). Thermal cycling conditions were optimized for each gene. For the *LPL* gene: 5 min initial denaturation at 95 °C; followed by 40 cycles of denaturation at 95 °C for 1 min, coupling at 54 °C for 1 min, and extension at 72 °C for 1 min; finally, extension at 72 °C for 5 min and indefinite holding at 4 °C. The *NPY* gene: initial denaturation at 95 °C for 5 min; 40 cycles of denaturation at 95 °C for 1 min, coupling at 65 °C for 1 min, and extension at 72 °C for 1 min; followed by a final extension at 72 °C for 5 min and indefinite holding at 4 °C. PCR products were confirmed by running them on 2% agarose gel electrophoresis after amplification (Appendix A). The amplified products were used to restrict enzyme digestion reactions, and were prepared with a total volume of 30 µL: 2 µL restriction buffer, 1 µL enzyme (*BglI* for *NPY*, *MspI* for *LPL*), 17 µL nuclease-free water, and 10 µL PCR product (Table 1). Reactions were incubated at 37 °C for 16 h and then stored at 4 °C. Digestion products were analyzed by 2% agarose gel electrophoresis to evaluate band patterns (Appendix A).

### 2.6. Statistical Analysis

Data analysis was performed using SPSS (Version 18.0 statistic software package). The distribution of DASS-21 subscale scores was evaluated using skewness and kurtosis values, the Kolmogorov–Smirnov test, and the Shapiro–Wilk test. The results of these tests were supported by skewness and kurtosis assessments. All tests indicated no significant deviation from normality (*p* > 0.05), with skewness and kurtosis values falling within the acceptable −2 to +2 range, confirming the suitability of applying parametric statistical analyses (Appendix A). Parametric tests (*t*-test, ANOVA) were applied to normally distributed data; non-parametric tests (Mann–Whitney *U*, Kruskal–Wallis) were used otherwise. Two-way ANOVA examined genotype × metabolic status interactions. Hardy–Weinberg equilibrium was checked for all SNPs. Statistical significance was set at *p* < 0.05.

## 3. Results

A detailed summary of the demographic characteristics of the 200 participants included in the study is provided in Table 2. The mean age did not differ significantly between men (50.4 ± 9.5 years) and women (48.8 ± 7.7 years), indicating that the sample was age-comparable across sexes. A significantly higher proportion of participants reported having attained higher education (*p* < 0.05). This trend aligns with the socioeconomic profile of the district in which the clinical center is located, where a substantial portion of residents are employed by the nearby university, thereby contributing to the predominance of individuals with advanced educational backgrounds in the study sample. Physical activity also showed a statistically significant difference between groups (*p* < 0.05). For the purpose of this study, physical activity was defined as engaging in at least 30 min of walking on two or more days per week. The observed variation suggests differing lifestyle patterns within the population and provides additional context for interpreting subsequent clinical and behavioral outcomes.

Furthermore, the Kruskal–Wallis test was used to examine differences in psychosomatic variables across various SNP groups. Among all tested associations, two variables—*LPL*-Depression and *LPL*-Anxiety—showed statistically significant differences, suggesting that *LPL* expression may vary between groups when stratified by depressive symptoms and anxiety levels. No significant differences were observed for the other psychosomatic indicators or for *NPY* expression (Table 3), indicating that these measures were largely consistent across genotypes. Interestingly, these findings imply a potential link between *LPL* regulation and specific psychosomatic outcomes, warranting further investigation to elucidate the underlying biological and psychological mechanisms.

To further explore these associations, a post hoc analysis was conducted to identify specific genotypes with significant differences. The results revealed a significant difference in *LPL*-Anxiety between the GA and GG genotypes (*p* = 0.01), whereas *LPL*-Depression did not show significant genotype-specific differences (*p* > 0.05) (Table 4). Descriptive characteristics of the *LPL*-Anxiety GA-GG comparison are presented in Table 5.

Subsequent ANOVA confirmed that there were no statistically significant differences among the studied variables across groups, with all *p*-values exceeding 0.05 (Table 6). These findings indicate that mean values were comparable and that within-group variability outweighed between-group differences, reinforcing the interpretation that both genetic factors and psychosomatic indicators remain largely consistent among participants. Collectively, these analyses provide a comprehensive overview of the relationships between SNP variations and psychosomatic measures, highlighting specific links worthy of further investigation.

## 4. Discussion

This study investigated the relationship between psychosocial factors, including stress, anxiety symptoms, and depressive symptoms, and the presence of MetS among adults residing in the Turkistan region of Kazakhstan. Recognizing the growing burden of both metabolic and mental health disorders in Central Asia, the research aimed to explore potential links between psychological well-being and metabolic risk in a community-based sample, as well as identify associated genetic factors. By examining these associations, the study sought to contribute to a better understanding of how psychosomatic indicators may influence the development of MetS within this population.

A total of 200 participants, of whom 82 had MetS, were recruited for this study. The results had a normal distribution, confirming the suitability of subsequent analysis, and showed a positive association between *LPL* rs326 gene polymorphism, anxiety, and participants with MetS, but not other psychometric dimensions. A study investigating genetic variants of *LPL* and MetS by Malik et al. reported that the G allele of single-nucleotide polymorphism (SNP)-rs326 was associated with increased BMI, and that it is an indication for an increase in obesity risk [23]. A systematic review and meta-analysis by Ji et al., 2023, claimed that the evidence from cross-sectional studies in which anxiety and MetS were analyzed as dependent variables consistently demonstrated a significant association between the two [24]. Similarly, analysis of a population-based retrospective cohort study of more than 300,000 participants illustrated that MetS significantly increased the risk of anxiety, which the authors suggested was due to the increase in the level of chronic inflammation [25].

While the relationship between MetS and anxiety is widely reported, Takeuchi and colleagues, who investigated the baseline metabolic syndrome and future anxiety in nearly 1000 Japanese males, reported that MetS did not increase the risk of anxiety [26]. A meta-analysis study by Ji and colleagues that pooled data from 24 cross-sectional studies reported a statistically significant association between anxiety and MetS (pooled OR of 1.14, 95% CI: 1.07–1.23) [24]. These conflicting findings likely reflect differences in how MetS and anxiety are defined and measured, the demographic and ethnic characteristics of study populations, and variation in study design and follow-up length. Indeed, diagnostic criteria, use of self-report scales vs. clinical interviews, and whether inflammation or other mediators are assessed can all influence results. Thus, further large-scale, prospective studies with standardized definitions and robust methodology are required to clarify whether, and under what conditions, MetS contributes to anxiety risk.

Animal and human studies have suggested a potential role for *NPY* in behavioral processes, and some have reported an association between *NPY* rs16147 polymorphism and metabolic disorders. For example, the rs16147 variant was reportedly associated with cardiovascular risk factors, adipokine levels, insulin resistance, and a higher prevalence of metabolic syndrome [27,28,29]. However, the present study did not observe any significant association between rs16147 and metabolic indicators (such as diabetes, high waist circumference, or elevated HDL levels) or psychosomatic indicators. This discrepancy could be due to genetic differences, phenotype definitions, statistical power, or gene-to-environment interactions. Thus, we could speculate that the effects of rs16147 are likely context-dependent, and hence, comparing the metabolic or psychiatric associations reported in other populations to the cohort used in this study might not be accurate.

The current work also showed no relationship between MetS and depression, a similar finding to a cross-sectional study conducted in Bangladesh, which reported no significant associations between MetS and depression or anxiety [30]. Equally, studies from Iran that investigated the relationship between MetS and depression reported no significant association between the two [31,32]. However, a systematic review and meta-analysis of various studies in China totaling nearly 18,000 participants showed an increase in the prevalence of depression among people with MetS compared to their healthy counterparts [33]. Likewise, Khanna et al., 2020, reported a strong association between depression and anxiety symptoms and body mass index [34]. The differences between these results are likely due to cultural differences. For instance, societies in Bangladesh and Iran have a positive view towards obesity, thus reducing the risk of anxiety or depression [30,32], which is likely the case in Kazakhstan as well.

## 5. Limitation

One important limitation of this study is its cross-sectional design, which precludes conclusions about causality between MetS and mental health outcomes.

Another limitation is due to the use of DASS-21, a self-reporting tool, making the results vulnerable to response biases such as social desirability or underreporting of symptoms. Moreover, DASS-21 is primarily designed to assess the severity of symptoms of depression, anxiety, and stress, rather than providing a clinical diagnosis. This may limit its ability to capture the full complexity of psychiatric conditions. Cultural and linguistic differences may also influence how respondents interpret and rate items, potentially affecting the cross-cultural validity of findings. Also, the sample was drawn from a single geographic region, which may restrict the generalizability of the findings to broader or more diverse populations. Finally, the relatively small sample size may have limited the statistical power to detect subtle associations and increased the risk of error. These factors highlight the need for future studies with larger, more representative, and longitudinal cohorts to validate and extend the present findings.

## 6. Conclusions

In conclusion, the present findings show that, except for the *LPL*-anxiety gene, psychosomatic indicators did not differ significantly when stratified by the presence of metabolic syndrome, gene type, or genotype. The results suggest that, within the studied population, variations in metabolic and genetic factors were not associated with measurable differences in stress, anxiety, or depressive symptoms. This outcome highlights the need for further research with larger and more diverse cohorts to clarify potential subtle interactions between psychometric health, metabolic status, and genetic variation.

## Figures and Tables

**Table 1 ijerph-22-01853-t001:** Primers and restriction enzymes used for *NPY* and *LPL* SNPs genotyping.

Gene	rs	Primer Sequence	Restriction Enzyme	Ampl (bp) Restriction	Products (bp)
** *NPY* **	rs16147	F: 5′-CGACTTAGGGAGCCACCCACACC-3′ R: 5′-CAGGTGCTTCCTACTCCGGCGCCCAG-3′	*BglI*	280	TT: 280 TC: 280 + 250 + 30 CC: 250 + 30
** *LPL* **	rs326	F: 5′-TACACTAGCAAT GTCTAGCTGA-3′ R: 5′-TCAGCTTTAGCCCAGAATGC-3′	MspI	488	AA: 488 GA: 488 + 317 + 171 GG: 317 + 171

**Table 2 ijerph-22-01853-t002:** Demographic data.

Variable	Number	Mean Age (years)	Healthy Participants	Participants with MetS	*p*-Value
Gender					
Men	50	50.4 ± 9.5	28	22	0.23
Women	150	48.8 ± 7.7	87	63	0.06
**Level of education ^#^**					
Higher education	168	48 ± 9.4	97	71	0.09
Middle School	32	54.7 ± 11.5	18	14	<0.05
**Employment**					
Employed	92	50.2 ± 12.8	60	32	0.31
Not employed	28	52.4 ± 14.6	12	16	<0.05
Self-employed	80	46.9 ± 10.1	43	37	0.07
**Physical activity ***					
Yes	177	49.1 ± 12.2	105	72	<0.05
No	23	49.6 ± 11.8	10	13	0.09

Data are presented as mean  ±  SD. ^#^ Higher education ≥ bachelor’s degree, Middle school ≤ 12 years of formal schooling; * Physical activity defined as 30 min of walking at least twice a week.

**Table 3 ijerph-22-01853-t003:** Kruskal–Wallis test results for non-parametric variables.

	Metabolic Syndrome	Gene	Psychosomatic Indicator	Chi-Square Value	Degrees of Freedom	*p*-Value
**1**	1	LPL	Anxiety	1.135659	2	0.56675
**2**	0	LPL	Depression	6.109565	2	0.04713
**3**	0	LPL	Anxiety	7.448795	2	0.02413
**4**	1	NPY	Anxiety	2.899900	2	0.23460
**5**	0	NPY	Anxiety	1.075245	2	0.58414

**Table 4 ijerph-22-01853-t004:** Post hoc analysis of *LPL*-Depression and *LPL*-Anxiety variables.

	Metabolic Syndrome	Gene	Psychosomatic Indicator	Comparison	*p*-Value
**1**	0	*LPL*	Depression	AA-GA	0.07452
**2**	0	*LPL*	Depression	AA-GG	0.49624
**3**	0	*LPL*	Depression	GA-GG	0.07084
**4**	0	*LPL*	Anxiety	AA-GA	0.18389
**5**	0	*LPL*	Anxiety	AA-GG	0.10763
**6**	0	*LPL*	Anxiety	GA-GG	0.01446 *

* designate statistical significance.

**Table 5 ijerph-22-01853-t005:** Description of LPL-Anxiety and GA-GG variables.

	Genotype	Number	Mean	Median	SD
**1**	GA	48	6.85	7	2.85
**2**	GG	8	3.87	2.5	4.12

**Table 6 ijerph-22-01853-t006:** Results of ANOVA test for variables with normal distribution.

	Metabolic Syndrome	Gene	Psychosomatic Indicator	Degrees of Freedom	Sum of Squares (Sum Sq)	Mean	F-Value	*p*-Value
**1**	1	*LPL*	Depression	2	9.73	4.86	0.4864349	0.6167
**2**	1	*LPL*	Stress	2	25.22	12.61	1.0173295	0.3663
**3**	1	*NPY*	Depression	2	21.93	10.96	1.1185652	0.3318
**4**	1	*NPY*	Stress	2	14.73	7.36	0.57664	0.5641
**5**	0	*LPL*	Stress	2	69.45	34.72	1.747375	0.1792
**6**	0	*NPY*	Depression	2	22.31	11.15	0.8620843	0.4252
**7**	0	*NPY*	Stress	2	8.83	4.41	0.2113154	0.8099

## Data Availability

The data presented in this study are available on request from the corresponding author due to privacy reasons.

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
