# Peer review of "Genetic Predictors of Psychosomatic Symptoms in Individuals with Metabolic Syndrome: Insights from a Cross-Sectional Study in Kazakhstan"

_ijerph, 2025, doi:10.3390/ijerph22121853_

Round 1

Reviewer 1 Report

Comments and Suggestions for Authors

The manuscript is current and well written. Besides that I noticed some errors that need to be corrected before publication.

  1. The objectives stated in the abstract (lines 10-12) are not consistent with the objectives stated at the end of the Introduction (lines 66-68). In the abstract it is stated that the aim of this study was to evaluate clinical and metabolic indicators of metabolic syndrome… What metabolic indicators are the authors referring to because it was not mentioned in the results.
  2. In the Discussion section (page 8, lines 219-220) it is stated that no significant association between rs16147 and metabolic indicators was observed. Please, specify which metabolic indicators this sentence refers to?
  3. I suggest the authors to expand and improve the Discussion section with more recent published articles and compare their study with other studies results, with a focus on what their study brought new in comparison to previous ones.
  4. By reviewing the Reference section, I noticed that more than 25% of cited papers were older than 15yrs. I suggest to refresh this section with more recent published articles.

Author Response

  1. The objectives stated in the abstract (lines 10-12) are not consistent with the objectives stated at the end of the Introduction (lines 66-68). In the abstract it is stated that the aim of this study was to evaluate clinical and metabolic indicators of metabolic syndrome… What metabolic indicators are the authors referring to because it was not mentioned in the results.

We thank the reviewer and apologize for this. We have rewritten the aims in the abstract.

  1. In the Discussion section (page 8, lines 219-220) it is stated that no significant association between rs16147 and metabolic indicators was observed. Please, specify which metabolic indicators this sentence refers to?

Response: we thank the reviewer for this comment. We have updated this sentence according to the reviewer’s suggestions in the discussion lines 241-242.

  1. I suggest the authors to expand and improve the Discussion section with more recent published articles and compare their study with other studies results, with a focus on what their study brought new in comparison to previous ones.

Response: We thank the reviewer for this point. The discussion has been updated and new reference have been added.

  1. By reviewing the Reference section, I noticed that more than 25% of cited papers were older than 15yrs. I suggest to refresh this section with more recent published articles.

 Response: we apologize for this. We have included new references.

Reviewer 2 Report

Comments and Suggestions for Authors

Dear Authors,

Thank you for your manuscript. I have provided some comments below that think could enhance the overall quality of your work.  

Kind Regards

Comments

Title:

  1. The phrase “Genetic Predictors of Psychosomatic…” is incomplete because psychosomatic is an adjective and needs a noun. You need to specify, e.g., psychosomatic symptoms (not functions as stated in the abstract). However, consider using different terminology to match the measures used, e.g., “psychosocial indicators,” “psychological indicators,” or “stress, anxiety, and depressive symptoms,” or another term that is scientifically more precise.

Abstract:

  1. Please consider using different terminology instead of “psychosomatic functions” or “psychosomatic indicators,” as mentioned above. If you choose to keep this term, please make sure you clearly explain it in the text and provide the necessary literature.
  2. There is a contradiction between the results and the conclusions. In the results section, you state that there is a significant difference in LPL–Anxiety between genotypes GA–GG (p < 0.05), but in the conclusion section this association is not mentioned.
  3. Please don’t use abbreviations without explaining them the first time they are mentioned, e.g., use “lipoprotein lipase gene (LPL)” instead of “LPL” alone.
  4. Please consider rewriting the abstract by starting the introduction section with a description of metabolic syndrome and then providing the aims in a clearer presentation. Please provide numbers in the results section and be more precise in the conclusions.
  5. Consider providing a graphical abstract.

Methods:

  1. Please define “metabolically healthy or unhealthy as diagnosed by health professionals.” What were the specific criteria for MetS?
  2. Are DASS-21 and PSQ validated for the population of Kazakhstan? If you used a previously validated version, please include a reference in the Methods. If they are not validated, the questionnaires must be at least translated and culturally adapted, and this should be acknowledged as a limitation.
  3. Please include a reference in the genotyping section. Has this protocol been published previously? Is it validated?

Results:

  1. Please present your results in a single paragraph without deviation according to the test used and be more explanatory about your tables. Please remove the bold letters. I recommend a major revision in the results section. 
  2. Please place the tables in the text close to their first citation, as stated in the instructions for authors: “All figures, schemes and tables should be inserted into the main text close to their first citation and must be numbered following their order of appearance.”

Author Response

Comments

Title:

  1. The phrase “Genetic Predictors of Psychosomatic…” is incomplete because psychosomatic is an adjective and needs a noun. You need to specify, e.g., psychosomatic symptoms (not functions as stated in the abstract). However, consider using different terminology to match the measures used, e.g., “psychosocial indicators,” “psychological indicators,” or “stress, anxiety, and depressive symptoms,” or another term that is scientifically more precise.

 Response: Thank you very much for the suggestion. The title has been changed according to the suggestion.

Abstract:

  1. Please consider using different terminology instead of “psychosomatic functions” or “psychosomatic indicators,” as mentioned above. If you choose to keep this term, please make sure you clearly explain it in the text and provide the necessary literature.

Response: Thank you very much for the suggestion. The term has been changed according to the suggestion.

  1. There is a contradiction between the results and the conclusions. In the results section, you state that there is a significant difference in LPL–Anxiety between genotypes GA–GG (p < 0.05), but in the conclusion section this association is not mentioned.

Response: we apologize for the confusion. We have included the change in the conclusion.

  1. Please don’t use abbreviations without explaining them the first time they are mentioned, e.g., use “lipoprotein lipase gene (LPL)” instead of “LPL” alone.

Response: WE apologize for this. The abbreviation is showing in the introduction line 42.

  1. Please consider rewriting the abstract by starting the introduction section with a description of metabolic syndrome and then providing the aims in a clearer presentation. Please provide numbers in the results section and be more precise in the conclusions.

Response: We thank the reviewer for this comment. We have started the abstract with a definition of MetS.

  1. Consider providing a graphical abstract.

 Response: We thank the reviewer for this comment. We have generated a graphical abstract. Please refer graphical abstract file.

Methods:

  1. Please define “metabolically healthy or unhealthy as diagnosed by health professionals.” What were the specific criteria for MetS?

Response: We apologize for omitting this. We have included the criteria for classification of metabolically health and unhealthy individuals.

  1. Are DASS-21 and PSQ validated for the population of Kazakhstan? If you used a previously validated version, please include a reference in the Methods. If they are not validated, the questionnaires must be at least translated and culturally adapted, and this should be acknowledged as a limitation.

Response: We thank the reviewer for this point. We have included a new subsection number 2.4, titled “Study validation and reproducibility” where we describe the steps we took to ensure questionnaire reproducibility.

  1. Please include a reference in the genotyping section. Has this protocol been published previously? Is it validated?

 Response: We thank the reviewer for this point. We have included a reference as suggested. Methods section 2.5- Genotyping.

Results:

  1. Please present your results in a single paragraph without deviation according to the test used and be more explanatory about your tables. Please remove the bold letters. I recommend a major revision in the results section. 

Response: We thank the reviewer for comment and agree with them. We have completely revised the results section and made it into one paragraph removing the subsection numbers.

  1. Please place the tables in the text close to their first citation, as stated in the instructions for authors: “All figures, schemes and tables should be inserted into the main text close to their first citation and must be numbered following their order of appearance.”

Response: We apologize for this. We have now changed the results section and included the tables as suggested

Reviewer 3 Report

Comments and Suggestions for Authors

In this study examine association between Psychosomatic indicators and Mets among 200 participants from Kazakhstan and if the genes paly a role in this association. They found association with anxiety but not depression. Overall study is good, but some sections need improvement especially methos and result section. Below are some suggestions for improvements.

Method

Need clearly specify inclusion.

What do you define individuals with MetS? This need clarification

Need to mention sociodemographic characteristics collected from participants

Line 128-129 test interaction on what needs to be specify

Linear regression analysis should be done for genotype 128 × metabolic status interactions.

Result

What do you mean by higher and middle education need to be more specific with wording use?

Asking participants if they are physical active (yes/No) is not valid and reliable measurement, thus it is better to delete this information

How is the study about Mets and in the beginning of introduction obesity and the data dose not have any information about body mass index of participants.

Add number of sample in each group in table 2 healthy participants and participants with MetS

Another point in table 2, categorical variables like gender, education, employment and physical activity should be presented as frequency (percentage). This table is completely wrong. What do you mean by number of samples in higher education is 168±10.8? This does not correct. This table needs to be reanalyzed

Normality assessment should be mentioned in brief in statistical section method not in the result and table 3 should move to supplementary materials because it is not part of the aim of study. It is just to check suitability of data for chosen test

In table 6 only 6 of participants have GG variant, usually in this case data should be combined between GA and GG

Table 4 and table 7 I do not How the same scale either depression or anxiety one time consider non-parametric (table 4) and another time (table 7) considered parametric. I think statistical analysis should be checked

Discussion

Line 197 the confirmed association between LPL and anxiety is it confirmed among healthy or participants with MetS.

Gene symbol should be written in italic check this through manuscript.

Author Response

Method

Need clearly specify inclusion.

Response: We apologize for this. Inclusion and exclusion criteria have been added in the methods section lines 82-85.

What do you define individuals with MetS? This need clarification

Response: We apologize for missing this point. We have included a clear definition about metabolicly health and unhealthy individuals. Methods section lines 75-82.

Need to mention sociodemographic characteristics collected from participants

Response: we thank the reviewer and agree that is important to mention the sociodemographic data in the methods section. We have mentioned the sociodemographic data to be collected in the methods section lines 85-87.

Line 128-129 test interaction on what needs to be specify

Response: We apologize for this. This section in the method has been rewritten, Line 150. 

Linear regression analysis should be done for genotype 128 × metabolic status interactions.

Response: We appreciate this point, however, the study focused on nonparametric and ANOVA-based comparisons. Also, the distribution characteristics of the genotype groups, limited the statistical power to reliably detect interaction effects.

Result

What do you mean by higher and middle education need to be more specific with wording use?

Response: We apologize for this. We have included an explanation in the footnote for the table.

Asking participants if they are physical active (yes/No) is not valid and reliable measurement, thus it is better to delete this information

Response: We apologize for this. The answers showing is a summary of several questions, but summarized in the table. Originally, the participants were asked whether they do any form of physical activities or walking, and if yes, is it more or less than 30 minutes, and then how many times per week. The results represent a summary. 

How is the study about Mets and in the beginning of introduction obesity and the data dose not have any information about body mass index of participants.

Response: we apologize for this. We have updated this sentence. Introduction section line 31.

Add number of sample in each group in table 2 healthy participants and participants with MetS

Response: we apologize for this. We have updated the table. Table 2.

Another point in table 2, categorical variables like gender, education, employment and physical activity should be presented as frequency (percentage). This table is completely wrong. What do you mean by number of samples in higher education is 168±10.8? This does not correct. This table needs to be reanalyzed.

Response: we apologize for this point. The table has been updated. Table 2 page 4-5.

Normality assessment should be mentioned in brief in statistical section method not in the result and table 3 should move to supplementary materials because it is not part of the aim of study. It is just to check suitability of data for chosen test

Response: we agree with the reviewer. Table 3 has been included in the statistical section and labeled as “supplementary table 1.”

In table 6 only 6 of participants have GG variant, usually in this case data should be combined between GA and GG

Response: We thank the reviewer for the suggestion. While it is common in some studies to combine GA and GG genotypes when the GG group is small, we chose to present them separately to highlight the distinct LPL-Anxiety values observed in the GG group, despite its limited size.

Table 4 and table 7 I do not How the same scale either depression or anxiety one time consider non-parametric (table 4) and another time (table 7) considered parametric. I think statistical analysis should be checked

Response: We thank the reviewer for their comment regarding the use of parametric and nonparametric analyses. In our study, the choice of statistical test was based on the distribution and group sizes of the variables. For Table 4, the distribution of the psychosomatic variables within specific genotype groups deviated from normality and/or had small group sizes, warranting the use of the nonparametric Kruskal–Wallis test. In contrast, for Table 7, the overall variables met normality assumptions and had sufficiently balanced group sizes, allowing for parametric ANOVA. We have clarified these criteria in the Methods section, statistical analysis subsection.

Discussion

Line 197 the confirmed association between LPL and anxiety is it confirmed among healthy or participants with MetS.

Response: we thank the reviewer and apologize for this. The sentence is corrected.

Gene symbol should be written in italic check this through manuscript.

Response: We thank the reviewer for this and apologize for missing this point. We have made all genes italic throughout the document.

Round 2

Reviewer 2 Report

Comments and Suggestions for Authors

Dear Authors,

Thank you for considering my comments and revising the manuscript accordingly. I appreciate your efforts to address the suggested improvements. In my opinion, the revised manuscript is scientifically sound. However, I was not able to locate the graphical abstract, and all graphs should be placed within the text. These issues can be corrected by the Academic Editor, and I am able to recommend your manuscript for publication.

Kind regards

Reviewer 3 Report

Comments and Suggestions for Authors

Thanks authors for addressing all raised comments, only one concerns regarding table 2 how categorical variables like education, employment presented in mean and standard division